# Cytochrome P450 1B1 Expression Regulates Intracellular Iron Levels and Oxidative Stress in the Retinal Endothelium

**DOI:** 10.3390/ijms24032420

**Published:** 2023-01-26

**Authors:** Yong-Seok Song, Ismail S. Zaitoun, Shoujian Wang, Soesiawati R. Darjatmoko, Christine M. Sorenson, Nader Sheibani

**Affiliations:** 1Department of Ophthalmology and Visual Sciences, University of Wisconsin School of Medicine and Public Health, Madison, WI 53705, USA; 2McPherson Eye Research Institute, University of Wisconsin School of Medicine and Public Health, Madison, WI 53705, USA; 3Department of Pediatrics, University of Wisconsin School of Medicine and Public Health, Madison, WI 53705, USA; 4Department of Cell and Regenerative Biology, University of Wisconsin School of Medicine and Public Health, Madison, WI 53705, USA; 5Department of Biomedical Engineering, University of Wisconsin, Madison, WI 53705, USA

**Keywords:** retinal vasculature, iron homeostasis, BMP6, hepcidin, ferroportin, eNOS, NF-κB, endothelial nitric oxide synthase

## Abstract

Cytochrome P450 (CYP) 1B1 is a heme-containing monooxygenase found mainly in extrahepatic tissues, including the retina. CYP1B1 substrates include exogenous aromatic hydrocarbons, such as dioxins, and endogenous bioactive compounds, including 17β-estradiol (E2) and arachidonic acid. The endogenous compounds and their metabolites are mediators of various cellular and physiological processes, suggesting that CYP1B1 activity is likely important in maintaining proper cellular and tissue functions. We previously demonstrated that lack of CYP1B1 expression and activity are associated with increased levels of reactive oxygen species and oxidative stress in the retinal vasculature and vascular cells, including retinal endothelial cells (ECs). However, the detailed mechanism(s) of how CYP1B1 activity modulates redox homeostasis remained unknown. We hypothesized that CYP1B1 metabolism of E2 affects bone morphogenic protein 6 (BMP6)-hepcidin-mediated iron homeostasis and lipid peroxidation impacting cellular redox state. Here, we demonstrate retinal EC prepared from *Cyp1b1*-deficient (*Cyp1b1^−/−^*) mice exhibits increased estrogen receptor-α (ERα) activity and expresses higher levels of BMP6. BMP6 is an inducer of the iron-regulatory hormone hepcidin in the endothelium. Increased hepcidin expression in *Cyp1b1^−/−^* retinal EC resulted in decreased levels of the iron exporter protein ferroportin and, as a result, increased intracellular iron accumulation. Removal of excess iron or antagonism of ERα in *Cyp1b1^−/−^* retinal EC was sufficient to mitigate increased lipid peroxidation and reduce oxidative stress. Suppression of lipid peroxidation and antagonism of ERα also restored ischemia-mediated retinal neovascularization in *Cyp1b1^−/−^* mice. Thus, CYP1B1 expression in retinal EC is important in the regulation of intracellular iron levels, with a significant impact on ocular redox homeostasis and oxidative stress through modulation of the ERα/BMP6/hepcidin axis.

## 1. Introduction

Cytochrome P450 1B1 (CYP1B1) is a member of the CYP450 family that catalyzes NADPH-supported mono-oxygenation of diverse molecules. It is one of the CYP enzymes found primarily in extrahepatic tissues including the retina [1]. CYP1B1 metabolizes xenobiotics, including polycyclic aromatic hydrocarbons. It also facilitates metabolism of physiologically reactive compounds, including 17β-estradiol (E2), polyunsaturated fatty acid (PUFA) products, retinol, and melatonin [2]. These signaling molecules are implicated in organ development and homeostasis, with strong CYP1B1 expression observed during murine [3] and human [4] eye development. In early 2000, mutations in the CYP1B1 gene were shown to be frequently associated with primary congenital glaucoma (PCG) in humans [5,6]. It is now well documented that CYP1B1 mutations are major genetic causes of PCG [7]. *Cyp1b1*-deficient (*Cyp1b1^−/−^*) mice exhibit developmental defects in the organization of their trabecular meshwork (TM) and Schlemm’s canal, as observed in humans with congenital glaucoma [8]. We showed TM cells and tissues prepared from *Cyp1b1^−/−^* mice exhibit increased oxidative stress [9]. TM cells share many characteristics with vascular cells [10,11]. CYP1B1 expression in vascular endothelial cells (ECs) is induced by shear stress [12], which impacts reactive oxygen species (ROS) production and oxidative stress [13]. Thus, CYP1B1 activity may play a key role in the regulation of cellular and tissue redox homeostasis. 

We previously reported that CYP1B1 expression and activity promote the proangiogenic activity of retinal EC in culture, and retinal neovascularization during oxygen-induced ischemic retinopathy (OIR) in vivo [14]. The retinas from *Cyp1b1^−/−^* mice subjected to OIR exhibited increased levels of lipid peroxidation and oxidative stress and mitigation of neovascularization. This was largely attributed to increased expression of thrombospondin-2 (TSP2), an endogenous inhibitor of angiogenesis [14]. Increased levels of lipid peroxidation products and oxidative stress were also observed in retinal EC isolated from *Cyp1b1^−/−^* mice, and in *Cyp1b1^+/+^* EC incubated with TMS (2,3′,4,5′-tetramethoxystilbene), a specific inhibitor of CYP1B1 activity [14]. *Cyp1b1^−/−^* retinal EC also showed sustained nuclear factor (NF)-κB activation [15]. Furthermore, expression of endothelium nitric oxide synthase (eNOS), the enzyme that produces nitric oxide (NO) in EC, was dramatically decreased in *Cyp1b1^−/−^* retinal EC [16]. NO serves many functions in EC. It acts as an anti-oxidant through the termination of lipid radical-mediated chain propagation reactions [17]. Collectively, these studies indicated that CYP1B1 deficiency in the EC was directly associated with increased oxidative stress levels. Unfortunately, how CYP1B1 expression impacts redox homeostasis and oxidative stress remains unresolved.

Previous studies have suggested roles for bioactive compounds metabolized by CYP1B1 in modulation of oxidative stress. CYP1B1 specifically catalyzes 4-hydroxylation of the aromatic ring of E2 to yield 4-hydroxyestradiol (4-OH-E2), which is followed by conjugative modification of 4-OH-E2 by uridine diphosphoglucuronosyl–transferases and their excretion [18]. In addition, mutations affecting CYP1B1 function could result in the accumulation of E2 [19]. E2 can induce bone morphogenic protein 6 (BMP6) expression through activation of estrogen receptor (ER)α [20]. Furthermore, BMP6 is recognized as a key mediator of hepcidin production and iron homeostasis [21]. We have recently shown that liver sinusoidal ECs express hepcidin in culture [22]. Hepcidin limits iron export by binding to the iron exporter ferroportin, promoting its internalization and degradation [23]. As ferroportin is the sole known cellular iron exporter [23], hepcidin and related signaling pathways are the key mechanisms in the regulation of systemic, and likely local, iron homeostasis in various tissues including the retina. The role of CYP1B1 in systemic and retinal iron homeostasis is unknown. 

Iron is an essential element, but in excess, it disrupts redox homeostasis and catalyzes the production and propagation of ROS leading to oxidative stress [24]. Specifically, iron is a potent generator of hydroxyl radicals, via the Fenton reaction, and is the most reactive among ROS. Furthermore, hydroxyl radicals and Fe^2+^ are molecules that initiate and propagate lipid peroxidation of polyunsaturated fatty acids, such as arachidonic acid (AA), to 4-hydroxyl-nonenal (4-HNE), a toxic lipid peroxidation product [25]. We previously reported that the lack of CYP1B1 results in increased levels of 4-HNE and sensitivity to oxidative stress in ocular tissues and cells [2,14,15]. These findings suggested a central role for CYP1B1 in the regulation of cellular redox state, most likely through modulation of iron homeostasis. The relationship among endothelium CYP1B1 expression, iron levels, and cellular redox homeostasis in the retina is the focus of studies presented here.

## 2. Results

### 2.1. Increased ERα Nuclear Localization and Expression of ERα Target Genes in Cyp1b1^−/−^ Retinal EC

Estrogen receptors (ERs) are members of the nuclear hormone receptor superfamily and act as ligand-activated transcription factors [26]. Two classes of ER have been identified: nuclear ERs, including ERα and ERβ, and membrane-bound ERs such as G-protein-coupled ER (GPER). Among nuclear ERs, ERα is more highly expressed than ERβ and is thought to be more important in mediating estrogen effects in the vascular endothelium [27]. To compare ERα levels in *Cyp1b1^+/+^* and *Cyp1b1^−/−^* retinal EC, we performed indirect immunofluorescence. Figure 1A shows that *Cyp1b1^−/−^* retinal ECs exhibit an increased nuclear localization of ERα compared to *Cyp1b1^+/+^* cells. The subcellular localization of ERα is predominately in the nucleus, both in the presence and absence of its ligand, E2 [28]. To test whether increased nuclear localization of ERα resulted in its transitivity, we measured the expression levels of its target genes, namely Gper [29,30] and estrogen-related receptor alpha (Esrra) [31]. The expression of both target genes was upregulated in *Cyp1b1^−/−^* retinal EC compared with *Cyp1b1^+/+^* retinal EC (Figure 1B). These data demonstrate that in the absence of CYP1B1 expression, retinal ECs show enhanced nuclear localization and transcriptional activity of ERα. 

### 2.2. Enhanced BMP6 Signaling in Cyp1b1^−/−^ Retinal ECs

17β-estradiol and its receptors modulate the expression of many genes, including BMP6, a member of the transforming growth factor-beta (TGFβ) family. Previous studies showed that incubation of cancer cells with E2 induces BMP6 expression [32] and BMP6 promoter–reporter construct activity, which is mainly mediated through ERα [20]. Here, we assessed BMP6 expression in the presence and absence of CYP1B1. Figure 2A shows that the BMP6 levels were upregulated in *Cyp1b1^−/−^* retinal ECs compared with *Cyp1b1^+/+^* retinal ECs.

We next examined the levels of proteins involved in the BMP6 signaling pathway via Western blot analysis. Both *Cyp1b1^+/+^* and *Cyp1b1^−/−^* retinal ECs expressed BMP receptor type II (BMPRII), the receptor to which BMP6 binds with high affinity [33]. SMAD1, 5, and 8 are transcription factors and the major intracellular mediators of BMP signaling [34], and phosphorylated by BMP receptors in a ligand-dependent manner, translocating into the nucleus [35]. *Cyp1b1^−/−^* retinal ECs exhibited enhanced SMAD1 phosphorylation levels compared with *Cyp1b1^+/+^* retinal ECs (Figure 2A). We also assessed whether E2 can induce BMP6 expression and/or activate its downstream signaling pathway in *Cyp1b1^+/+^* retinal ECs. *Cyp1b1^+/+^* and *Cyp1b1^−/−^* retinal ECs were incubated with E2 (1 or 10 nM) for 24 h and cell lysates were prepared for Western blot analysis. Figure 2B shows incubation of retinal ECs with E2 increased BMP6 protein levels and SMAD1 phosphorylation in *Cyp1b1^+/+^* ECs, while the basal levels of these proteins were elevated in *Cyp1b1^−/−^* ECs with or without E2. Further, we performed indirect immunofluorescence staining to determine the cellular localization of phosphorylated SMAD1 (p-SMAD1) and total SMAD1. As shown in Figure 2C,D, pSMAD1 nuclear localization was enhanced in *Cyp1b1^−/−^* and *Cyp1b1^+/+^* retinal ECs. 

### 2.3. Increased Intracellular Fe^2+^ Levels in Cyp1b1^−/−^ Retinal ECs

The BMP6-SMAD signaling pathway has an important role in regulating hepcidin expression in the endothelium. Hepcidin is a peptide hormone that binds to the cellular iron exporter ferroportin, which induces internalization and degradation of ferroportin, thus, decreasing cellular iron export and increasing cellular iron retention [36]. Consistent with the induced BMP6 levels, *Cyp1b1^−/−^* retinal ECs showed increased Hamp (hepcidin gene) expression compared to *Cyp1b1^+/+^* retinal ECs. In addition, Slc40a1 (ferroportin gene) expression was downregulated in *Cyp1b1^−/−^* retinal ECs (Figure 3A). These results demonstrated that CYP1B1 expression regulates intracellular iron levels, and its absence in retinal ECs affects intracellular iron levels.

Next, intracellular iron levels in *Cyp1b1^+/+^* and *Cyp1b1^−/−^* retinal ECs were assessed by staining ECs with the Fe^2+^-selective fluorescent probe FerroOrange. Here, we showed increased FerroOrange staining intensity in *Cyp1b1^−/−^* retinal ECs compared with *Cyp1b1^+/+^* retinal ECs. Incubation with the iron chelator deferoxamine (DFO) (10 µM, 48 h) suppressed the FerroOrange staining in *Cyp1b1^−/−^* retinal ECs. In addition to suppressing intracellular iron levels, DFO incubation (10 µM DFO for 24 h) also altered EC morphology from an elongated and spindly morphology in *Cyp1b1^−/−^* retinal ECs to that very similar to *Cyp1b1^+/+^* retinal EC morphology (Figure 3C). 

### 2.4. Removal of Excess Iron Reverses the Cellular Changes in Cyp1b1^−/−^ Retinal ECs

We previously reported that *Cyp1b1^−/−^* retinal ECs have sustained NF-κB activation with increased phosphorylated p65 (pp65) levels [15]. These cells also exhibited decreased eNOS expression compared with *Cyp1b1^+/+^* retinal ECs [14]. We next used the iron chelator DFO to examine whether iron excess was responsible for the cellular changes noted in *Cyp1b1^−/−^* retinal ECs. Cells were incubated with 10 or 20 µM DFO for 24 h. Western blot analysis showed that the removal of excess iron mitigated the increase in pp65 levels (Figure 4A) and increased eNOS expression in *Cyp1b1^−/−^* retinal ECs (Figure 4B).

HIF-1 is a heterodimeric transcription factor composed of HIF-1α and HIF-1β. The stability and activity of HIF-1α are modulated by post-translational modifications, including hydroxylation by prolyl hydroxylase (PHD) proteins. PHDs are members of the 2-oxoglutarate/Fe^2+^-dependent dioxygenase and hydroxylate prolyl residues in HIF-1α, which is followed by von Hippel–Lindau protein-mediated ubiquitination and further 26 S proteasomal degradation [37]. Thus, altered iron homeostasis could impact the post-translational modification of HIF-1. For example, iron supplementation (40 µM of FeCl_2_) dramatically reduces HIF-1α levels in cancer cells in vitro [38]. Consistent with increased intracellular iron levels, *Cyp1b1^−/−^* retinal ECs showed lower HIF-1α expression compared with *Cyp1b1^+/+^* retinal ECs. Incubation of *Cyp1b1^−/−^* retinal ECs with DFO (50 µM for 24 h) restored HIF-1α levels. DFO incubation had a minimal effect on HIF-1α levels in *Cyp1b1^+/+^* retinal ECs (Figure 4C). 

BMP6 is reported to increase vascular permeability and induce the internalization of VE cadherin, leading to a disruption in adheren junctions in vitro [39]. To determine VE-cadherin expression in *Cyp1b1^+/+^* and *Cyp1b1^−/−^* retinal ECs, lysates from confluent cells were prepared and VE-cadherin levels were assessed by Western blot analysis. Confluent *Cyp1b1^+/+^* retinal ECs, either incubated with vehicle or DFO, expressed significant levels of VE cadherin (Figure 4D). In contrast, confluent *Cyp1b1^−/−^* retinal ECs lacked detectable VE-cadherin levels (Figure 4D). Incubating *Cyp1b1^−/−^* retinal ECs with DFO (10 or 20 µM for 24 h) restored VE cadherin to detectable levels. These data collectively indicate that excess intracellular iron levels are responsible for the morphological and biochemical changes noted in *Cyp1b1^−/−^* retinal ECs.

As shown in Figure 5A, *Cyp1b1^−/−^* retinal EC failed to form a closely apposed monolayer of cells at confluence and lacked detectable VE-cadherin junctional localization. Incubation with DFO (20 µM for 48 h), the spindly morphology of confluent *Cyp1b1^−/−^* retinal ECs, restored VE-cadherin expression and junctional localization (Figure 5A). To delineate the mechanism of iron chelation and the morphological changes in *Cyp1b1^−/−^* retinal ECs, we tested two additional iron chelators, namely deferiprone (DFP, membrane-permeable Fe^3+^ chelator) and 2,2’-Bipyridine (BIP, membrane-permeable Fe^2+^ chelator). Neither of these two iron chelators affected the morphology of confluent *Cyp1b1^−/−^* retinal ECs (Figure 5B). DFO is a nonmembrane-permeable Fe^3+^ chelator that is taken up into the cells by endocytosis and, thus, accumulates in lysosomes where DFO chelates iron [40]. Thus, these data indicate that CYP1B1 expression is important in iron homeostasis, especially in the lysosomes.

### 2.5. Enhanced Lipid Peroxidation in Cyp1b1^−/−^ Retinal ECs

Iron is an essential element but, in excess, could disrupt redox homeostasis by promoting hydroxyl radical generation via Fenton chemistry and, subsequently, lipid peroxidation [41]. We examined lipid peroxidation levels via indirect immunofluorescence staining of *Cyp1b1^+/+^* and *Cyp1b1^−/−^* retinal ECs using antibodies for a lipid peroxidation product, 4-hydroxylnonenal (4-HNE). *Cyp1b1^−/−^* retinal ECs showed increased levels of 4-HNE staining compared with *Cyp1b1^+/+^* retinal ECs. Incubation with DFO (50 µM for 24 h) was sufficient to reduce 4-HNE levels in *Cyp1b1^−/−^* retinal ECs (Figure 6A,B). We also assessed acrolein levels, another lipid peroxidation product, via indirect immunofluorescence staining. *Cyp1b1^−/−^* retinal ECs exhibited higher intracellular acrolein levels compared with *Cyp1b1^+/+^* retinal ECs (Figure 6C). The increased acrolein levels in *Cyp1b1^−/−^* retinal ECs were also suppressed by DFO incubation (10 µM for 24 h; Figure 6D). Although DFO incubation also decreased acrolein levels in *Cyp1b1^+/+^* retinal ECs, the effect did not reach the significance level (*p* = 0.4138). 

### 2.6. Increased Ferroptosis Sensitivity in Cyp1b1^−/−^ Retinal ECs Is Associated with ERα Activity

The increased cellular iron level and lipid peroxidation in *Cyp1b1^−/−^* retinal ECs suggest an enhanced rate of ferroptosis in these cells. Ferroptosis is a form of iron-dependent cell death characterized by excessive accumulation of lipid peroxides and ROS [42]. To compare ferroptosis sensitivity of *Cyp1b1^+/+^* and *Cyp1b1^−/−^* retinal ECs, we incubated the cells with different concentrations (0.05–10 µM) of erastin for 24 h. Erastin induces ferroptosis through suppression of cysteine uptake via system X_c_^−^ [43]. Cell viability in response to erastin was analyzed via MTS assay and the results showed that erastin decreased cell viability in a dose-dependent manner. However, *Cyp1b1^−/−^* retinal ECs were more sensitive to Erastin (0.5 µM) incubation. Erastin reduced the viability of *Cyp1b1^−/−^* retinal ECs by 67.4%, while that of *Cyp1b1^+/+^* retinal ECs was decreased by 11.2% (Figure 7A,B). Thus, *Cyp1b1^−/−^* retinal ECs were more susceptible to erastin-mediated ferroptosis compared with *Cyp1b1^+/+^* retinal ECs. 

We next determined the effect of ERα antagonism on erastin-induced ferroptosis. First, we assessed the concentration of methyl-piperidino-pyrazole (MPP), a highly selective ERα antagonist [44], by incubating the cells with different concentrations of MPP (1–20 µM) for 48 h. MPP at 1 µM did not affect cell viability, but 5 µM of MPP reduced cell viability by 10% in both *Cyp1b1^+/+^* and *Cyp1b1^−/−^* retinal ECs. MPP at 10 µM reduced the viability of *Cyp1b1^+/+^* and *Cyp1b1^−/−^* retinal ECs by 50% and 60%, respectively (data not shown). Here, we chose to incubate cells with 0.5–5 µM of MPP for 24 h and then 0.5 µM erastin for an additional 24 h, following which viability was assessed with the MTS assay. As shown in Figure 7C,D, MPP incubation protected *Cyp1b1^−/−^* retinal ECs from erastin-induced ferroptosis but had a minimal impact on the viability of erastin-stimulated *Cyp1b1^+/+^* retinal ECs. These data suggest that CYP1B1 expression is important in the regulation of iron levels and lipid peroxidation, which drives ferroptosis sensitivity [45] through enhanced ERα activity. 

### 2.7. Restoration of Retinal Neovascularization in Cyp1b1^−/−^ Mice by ERα Antagonist

We previously reported the attenuation retinal neovascularization in *Cyp1b1^−/−^* mice during OIR [14]. CYP1B1 plays important roles in the modulation of ERα activity in retinal ECs, which impacts iron homeostasis and lipid peroxidation. Based on these observations, we next assessed the role ERα signaling plays in retinal neovascularization during OIR by administrating MPP to *Cyp1b1^+/+^* and *Cyp1b1^−/−^* mice (1 mg/kg, IP injection, prepared in 50 µL of saline), once the mice were returned to room air (P12) until P17 (maximum neovascularization) [46]. Retinas from P17 mice were wholemount immunostained with anti-collagen IV. Retinal neovascularization was assessed by measuring the area of neovascular tufts and calculated as a percentage of the total retinal area. Consistent with our previous studies [14,47], P17 male and female *Cyp1b1^−/−^* mice showed reduced neovascularization compared to P17 male and female *Cyp1b1^+/+^* mice (Figure 8A,C). We also found that female *Cyp1b1^−/−^* mice exhibit significantly less neovascularization than male *Cyp1b1^−/−^* mice at P17 (Figure 8C). ERα antagonism by MPP did not significantly affect retinal neovascularization in *Cyp1b1^+/+^* mice (Figure 8B). However, MPP injection did restore retinal neovascularization in P17 female, but not in male, *Cyp1b1^−/−^* mice (Figure 8D).

### 2.8. Lipid Peroxide Chelation Restored Retinal Neovascularization in Cyp1b1^−/−^ Mice 

We previously showed that the attenuation of retinal neovascularization in the absence of CYP1B1 was restored by administration of antioxidant NAC. These results suggested that CYP1B1 plays an important role in the regulation of oxidative stress during OIR, as demonstrated by the increased 4-HNE levels in retinas from *Cyp1b1^−/−^* mice during OIR [14]. To determine whether suppression of lipid peroxidation impacts retinal neovascularization, *Cyp1b1^−/−^* mice were administrated with ferrostatin-1 (Fer-1; 5 mg/kg from P12-P17), a synthetic antioxidant that scavenges alkoxyl radicals and inhibits lipid peroxidation [48]. Inhibition of lipid peroxidation by Fer-1 significantly restored neovascularization in P17 female, but not male, *Cyp1b1^−/−^* mice (Figure 9A,B). 

### 2.9. Effects of Iron Chelators on Retinal Neovascularization

Previous studies demonstrated that iron chelation suppresses oxidative stress in vivo. DFO reduces superoxide production and NADPH oxidase activity in fat tissues of diabetic mice [49]. Further, DFO administration suppresses levels of malondialdehyde, a lipid peroxidation product, induced by peripheral surgical brain trauma in mice [50]. Intraperitoneal administration of DFO also protects the photoreceptors of albino rats from intense light damage [51]. To determine whether iron chelation with DFO could impact retinal neovascularization, *Cyp1b1^−/−^* mice were administered DFO daily upon return to room air following hyperoxia. DFO administration modestly reduced neovascularization in both male and female *Cyp1b1^−/−^* mice, but the differences did not reach significant levels compared to the control groups (Appendix A; (*p* = 0.1264 in male mice and *p* = 0.2178 in female mice)).

To further delineate the effect of iron chelation on retinal neovascularization, we used the iron chelator DFP. DFP and DFO are both Fe^3+^ chelators, but DFO is a randomly oriented linear molecule with a low membrane permeability [52]. DFP is a small hydroxypyridinone molecule with high membrane permeability (Appendix A), which suggests that DFP has the potential to be more effective than DFO in chelating iron [53]. DFP protects retinal degeneration in mice mediated by sodium iodate and intense light and is neuroprotective in a preclinical mouse model of glaucoma [54,55,56]. *Cyp1b1^+/+^* and *Cyp1b1^−/−^* mice were intraperitoneally administrated with 50 mg/kg or 100 mg/kg DFP every day from P12 to P17 during OIR. In *Cyp1b1^+/+^* mice, 100 mg/kg DFP significantly downregulated neovascularization in male and female mice, but the effect of 50 mg/kg DFP in *Cyp1b1^+/+^* mice was not significant (Figure 10). However, DFP significantly decreased neovascularization in both male and female *Cyp1b1^−/−^* mice, at both 50 and 100 mg/kg (Figure 11). 

### 2.10. Iron Supplementation Using Ferric Ammonium Citrate (FAC) Did Not Alter Retinal Neovascularization in Cyp1b1^+/+^ and Cyp1b1^−/−^ Mice during OIR

We next assessed whether iron supplementation impacts retinal neovascularization during OIR. Ferric ammonium citrate (FAC) is a form of iron that has been widely used as an iron supplement [57]. Multiple in vitro studies have used FAC as an iron donor [58]. In adult mice, FAC (20 mg/kg, every-other-day IP injection for 2 weeks) suppressed vascular endothelial growth factor (VEGF) and tumor-cell-induced angiogenesis in vivo [59]. Male and female *Cyp1b1^+/+^* and *Cyp1b1^−/−^* mice were administered with 20 mg/kg FAC every day from P12 to P17 following hyperoxia. FAC administration did not significantly impact neovascularization in either male or female *Cyp1b1^+/+^* and *Cyp1b1^−/−^* mice (Appendix A).

## 3. Discussion

Here, using *Cyp1b1^+/+^* and *Cyp1b1^−/−^* retinal ECs, we demonstrate that CYP1B1 deficiency leads to a significant increase in ERα activity, upregulation of BMP6 expression, and activation of its downstream signaling mediator, suppressor of Mothers Against Decapentaplegic (SMAD) proteins. We also showed that incubation of *Cyp1b1^+/+^* retinal ECs with E2 induced the BMP6 signaling pathway. Enhanced BMP6 signaling in ECs induces hepcidin expression, suppressing the level of cellular iron exporter ferroportin, resulting in intracellular iron accumulation. Retinal ECs are the key cell type involved in forming the inner retinal blood barrier and are a major site for the regulation of retinal iron homeostasis [60]. Using an iron-sensing molecular probe, FerroOrange, we observed increased intracellular iron levels in *Cyp1b1^−/−^* retinal ECs compared to *Cyp1b1^+/+^* retinal ECs. Removal of excess iron reversed the phenotypic changes noted in *Cyp1b1^−/−^* retinal ECs, including sustained NF-κB p65 phosphorylation, diminished junctional localization of VE cadherin, decreased hypoxia-induced factor (HIF)-1α levels, and enhanced lipid peroxidation. These data suggested that *Cyp1b1^−/−^* retinal ECs should be more prone to ferroptosis, an iron-dependent cell death. We showed that *Cyp1b1^−/−^* retinal ECs have increased sensitivity to erastin-induced ferroptosis. In addition, removal of excess iron or ERα antagonism mitigated erastin-induced ferroptosis in *Cyp1b1^−/−^* retinal ECs. Removal of excess iron and ERα antagonism also restored retinal neovascularization in *Cyp1b1^−/−^* mice during OIR. Collectively, these data establish an important role for CYP1B1 expression in the regulation of retinal iron homeostasis and oxidative stress, through modulation of ERα activity and the BMP6–hepcidin axis in the retinal endothelium.

CYP1B1 is a key enzyme in the metabolism of E2 [61]. Loss of CYP1B1 function due to mutations results in reduced E2 metabolites, and increased E2 accumulation and estrogen receptor (ER) activation [19]. Two types of ER have been described, the nuclear receptors (ERα and ERβ) and the membrane receptors, such as GPER/GPR30. Ligand-bound ERα and ERβ undergo conformational changes including phosphorylation [62]. The activated ER then recruits two steroid receptor coactivator 3 (SRC-3) proteins that, in turn, bind to p300 to form an active complex on the estrogen response element (ERE) to regulate gene expression [63]. GEPR/GRP30 is located in both the cell membrane and the endoplasmic reticulum. The expression pattern of each ER is tissue and cell specific. Retina expresses ERα, ERβ, and GPER/GPR30 [64,65,66], and the expression of ERα and ERβ in retinal ECs has been previously demonstrated [67]. However, the impact of E2-ER signaling in retinal vascular function needs further investigation. Here, we showed that ERα nuclear localization is enhanced in *Cyp1b1^−/−^* retinal ECs, which was further confirmed by increased expression of ER-regulated target genes, including Esrra and Gper. Further, CYP1B1 expression is normally induced by E2. When activated by its ligand, ER binds to a putative ERE in the CYP1B1 promoter region (between −84 and −49) and induces CYP1B1 expression [68]. Thus, CYP1B1 plays an important role in ER activation and downstream signaling pathways through modulation of E2 levels.

BMP6 is a member of the transforming growth factor beta superfamily produced by mammalian oocytes and other cell types [69]. BMP6 has been investigated as a mediator of E2-dependent osteogenesis, where E2 injection in mouse long bones enhanced BMP6 expression [70]. However, accumulating evidence suggests tissue-selective effects of E2-ER signaling on BMP6 regulation. For example, E2 induces BMP6 in human osteoblastic cells [71], ER-positive breast cancer cells [72], and hepatocytes [32], but not in mesenchymal cells [73]. Furthermore, recent studies have demonstrated that BMP6 is a key endogenous molecule responsible for hepcidin expression by EC. However, the roles of E2-ER signaling in systemic (liver) and local (retina) regulation of iron homeostasis are not fully appreciated. For instance, studies have shown that E2 treatment could result in downregulation [74] or upregulation [32] of hepatocytic hepcidin. In addition, it is thought that hepatocytes in the liver are the source of hepcidin, which is driven by paracrine action of BMP6 produced by liver ECs. We have observed decreased hepcidin expression in the liver and liver ECs from *Cyp1b1^−/−^* mice [22]. However, the relationship between E2 metabolism by CYP1B1 and BMP6–hepcidin expression in tissues such as the retina remains to be further defined. In the studies presented here, *Cyp1b1^−/−^* retinal ECs showed increased BMP6 expression and upregulated phosphorylation of BMP6 downstream mediator SMADs. Furthermore, incubation of *Cyp1b1^+/+^* retinal ECs with E2 was sufficient to induce the BMP6–SMAD signaling axis. 

Separated from systemic circulation by the blood–retinal barrier, in the retina, iron homeostasis is locally achieved by interactions between iron-regulatory proteins produced locally. In fact, the retinal pigment epithelium, the cell layer that forms the outer blood–retinal barrier, expresses iron-regulatory genes, including hemochromatosis (Hfe), hemojuvelin (Hjv), transferrin receptor (Tfrc), Slc40a1 (gene encoding ferroportin), and Hamp (gene encoding hepcidin) [75]. Retinal ECs, as the key part of the inner blood–retinal barrier, play important roles in the regulation of retinal iron homeostasis. Retinal ECs express transferrin receptor 1 on their apical side for cellular iron import. Further, ferroportin expression in retinal ECs was found on the abluminal (basolateral) membrane facing the neuroretina, indicating that retinal ECs are responsible for importing systemic iron into the retina [76]. Activation of the BMP6–SMAD signaling pathway induces the expression of hepcidin, a peptide hormone that blocks ferroportin and induces the internalization and degradation of ferroportin [36]. Here, we showed that increased BMP6 activity in *Cyp1b1^−/−^* retinal ECs resulted in increased hepcidin expression and decreased ferroportin expression. These results demonstrate an autocrine BMP6–hepcidin signaling axis in retinal ECs. 

Increased hepcidin expression and decreased ferroportin expression can lead to cellular iron retention. Iron is an essential element for life, but when in excess, it takes part in the Fenton reaction, generating hydroxyl radicals, which are highly reactive and initiate lipid peroxidation [77]. Thus, iron homeostasis has the potential to play a key role in regulation of cellular redox state and cell function. *Cyp1b1^−/−^* retinal ECs showed that increased intracellular iron levels and iron chelation, for the most part, reversed the cellular changes noted in these cells. Iron chelation mitigated sustained NF-κB p65 phosphorylation and restored eNOS expression. In addition, iron chelation restored intracellular HIF-1α protein levels and junctional VE-cadherin expression. Furthermore, we showed that *Cyp1b1^−/−^* retinal ECs exhibit increased intracellular levels of lipid peroxidation products 4-HNE and acrolein that were also suppressed by iron chelation.

Recently termed, ferroptosis is an iron-dependent non-apoptotic form of cell death [78]. Underlying mechanisms, such as the inhibition of glutathione peroxidase 4 (GPX4) induced by GSH depletion and accumulation of redox-active Fe^2+^ and ROS, can induce ferroptosis. Excessive lipid peroxidation is the leading cause of cell death by ferroptosis [79]. Thus, oxidation of phospholipids, such as phosphatidylethanolamines, harboring AA is a critical molecular mechanism of ferroptosis [80]. Previous studies by our group demonstrated the regulation of oxidative stress by CYP1B1 expression in retinal ECs [14,15,16]. However, the underlying mechanisms responsible for this oxidative stress, and the role CYP1B1 plays in ferroptosis regulation remained unknown. Since the retinal vasculature in vivo and isolated retinal ECs from *Cyp1b1^−/−^* mice do not show increased cell death under normal conditions [14], it is likely that the absence of *Cyp1b1* expression increases oxidative stress to an extent that is not sufficient to drive cell death. *Cyp1b1^−/−^* retinal ECs can be more susceptible to oxidative damage, such as suppression of glutathione production, by the ferroptosis inducer erastin, a chemical antagonist of cysteine transporter system X_c_^−^ [43]. We showed that *Cyp1b1^−/−^* retinal ECs are, indeed, more prone to erastin-induced ferroptosis. Furthermore, antagonism of ERα by MPP mitigated erastin-induced ferroptosis in *Cyp1b1^−/−^* retinal ECs. Thus, these findings support an important role for CYP1B1 regulation of ERα activity and modulation of oxidative stress driven by an iron-dependent lipid peroxidation in retinal ECs and the retinal vasculature.

Retinal vascular diseases, including diabetic retinopathy and retinal vein occlusion, are the main causes of vision loss in developed countries [81]. Oxidative stress is recognized as a key contributor to the generation and progression of many retinal vascular diseases [82,83]. The molecules involved in redox homeostasis have been studied, and accumulating evidence from our studies suggests that CYP1B1 plays an important role in the regulation of ocular redox homeostasis. Our in vitro studies showed elevated intracellular iron levels mediated through ERα activity enhanced the BMP6–hepcidin signaling axis in *Cyp1b1^−/−^* retinal ECs, resulting in intracellular iron accumulation and increased oxidative stress. We extended these studies in vivo, examining retinal angiogenesis during OIR and showed attenuation of retinal neovascularization in *Cyp1b1^−/−^* mice was largely attributed to the elevated levels of oxidative stress [14].

Previous studies have shown a protective effect of E2 on oxidative stress during the early hyperoxic phase of OIR. E2 administration (daily subcutaneous injection from P7) reduced avascular area at P9 and downregulated malondialdehyde levels, an indicator of fatty acid oxidation, at P9 and P13 during OIR [84]. However, the roles of E2 in OIR may vary under different oxygen status, as the same group previously showed that E2 administration during hyperoxia (P7 to P11, daily subcutaneous injection) increased avascular area at P17 during OIR. In addition, E2 injection during the second phase of OIR (P12 to P16, daily subcutaneous injection) reduced neovascularization at P17 [85]. Our studies showed that ERα antagonist MPP administration during the second phase of OIR (P12-P17) did not affect retinal neovascularization of male and female *Cyp1b1^+/+^* mice. However, MPP treatment restored neovascularization more effectively in female *Cyp1b1^−/−^* mice. These results suggest that the E2 level is increased in the absence of CYP1B1, suppressing neovascularization through increased oxidative stress. However, the roles of estrogen signaling and the mechanism of E2 action on retinal angiogenesis in vivo deserve further elucidation. 

We previously demonstrated that during OIR, P17 *Cyp1b1^−/−^* mice have increased levels of 4-HNE, a toxic lipid peroxidation product, in their retina compared with P17 *Cyp1b1^+/+^* mice. Due to its highly reactive electrophilicity and membrane permeability, 4-HNE is regarded as a cytotoxic compound that can covalently modify any protein in the cytoplasm and nucleus [25]. We found systemic chelation of alkoxyl radicals by Fer-1 administration restored neovascularization in female *Cyp1b1^−/−^* mice, but not in male *Cyp1b1^−/−^* mice. 4-HNE is produced from AA by non-enzymatic reactions. ROS, such as hydroxyl radical (OH^•^), promote the abstraction of H from a methylene group (-CH_2_-) of PUFAs. The reaction leaves an unpaired electron on the carbon, and it is followed by molecular rearrangements to yield a lipid hydroperoxide. Lipid hydroperoxides undergo decomposition catalyzed by reduced transition metals including Fe^2+^. As a result of these reactions, lipid-peroxidation-derived aldehydes such as 4-HNE are generated [86]. Thus, iron has a key role in the formation of non-enzymatic 4-HNE production as well as the Fenton reaction to yield OH^•^. In fact, in aging eyes, excess iron accumulation is linked to retinal cell death and degeneration [87].

To determine the effect of iron chelation on retinal neovascularization during OIR, we used two FDA-approved iron chelators, DFO and DFP [88]. We hypothesized that iron chelation could alleviate oxidative stress and restore retinal neovascularization in *Cyp1b1^−/−^* mice, as we showed previously with NAC administration [14]. Here, we show that systemic iron chelation by DFP, but not by DFO, significantly downregulated retinal neovascularization in *Cyp1b1^+/+^* and *Cyp1b1^−/−^* mice. This may be due to the chemical and soluble nature of these chelators. Thus, removal of excess iron during the ischemic phase of OIR failed to restore retinal neovascularization in *Cyp1b1^−/−^* mice to the levels seen in *Cyp1b1^+/+^* mice. This could be attributed to the timing, duration, and/or amounts of chelators administered and deserves further investigation. In addition, iron supplementation during the ischemic phase of OIR did not impact retinal neovascularization in either *Cyp1b1^+/+^* or *Cyp1b1^−/−^* mice. To our knowledge, this is the first report of the administration of an iron chelator during the ischemic phase of OIR suppressing retinal neovascularization. However, the effect of DFP on mitigation of neovascularization was more effective in *Cyp1b1^−/−^* mice than *Cyp1b1^+/+^* mice, perhaps due to differences in systemic iron level changes in *Cyp1b1^−/−^* mice compared to *Cyp1b1^+/+^* mice. 

The studies presented here demonstrated a sex impact on the outcomes of iron chelation studies. Previous studies have indicated sex differences in iron levels in humans and various mouse strains. However, the mouse strain used in the studies presented here (C57BL/6J) did not show sex differences in retinal iron levels [89,90]. Thus, the reason for sex differences noted in our studies is unclear and may be impacted by the levels of E2 and ERα expression and activity. In addition, differences in liver vs. retina iron levels, in *Cyp1b1^−/−^* mice, may contribute to sex differences noted here. This notion is supported by a study showing that hepcidin deletion in the liver, and not in the retina, results in retinal increased iron levels and degeneration [91]. In addition, short duration of treatment and oxygen levels could also contribute to sex-dependent responses noted here. Thus, the roles of CYP1B1 in the regulation of systemic iron homeostasis and its impact on retinal local iron levels deserve further investigation.

In summary, we demonstrated that ERα activity and the BMP6–hepcidin axis are induced in the *Cyp1b1^−/−^* retinal ECs. The induction of the BMP6–hepcidin axis acting on iron exporter ferroportin in *Cyp1b1^−/−^* retinal ECs resulted in elevated intracellular iron levels and increased lipid peroxidation. In addition, iron chelation reduced sustained NF-κB p65 activation and lipid peroxidation, restoring junctional VE-cadherin expression and HIF-1α and eNOS levels in *Cyp1b1^−/−^* retinal ECs in vitro and neovascularization in vivo. Concomitant with upregulated cellular iron and lipid peroxidation levels, *Cyp1b1^−/−^* retinal ECs showed increased erastin-induced ferroptosis, which was mitigated by ERα antagonism. Together, our results indicate that CYP1B1 expression is important for maintaining the local cellular redox state through modulation of local iron homeostasis via the ERα–BMP6-hepxidin axis. Thus, CYP1B1 activity keeps oxidative stress in check through regulation of ER activity. Modulation of CYP1B1 activity may provide a suitable target to promote or inhibit neovascularization. Investigating the detailed molecular mechanisms involved in CYP1B1 metabolism of E2 and its other substrates will provide additional insight into the role CYP1B1 activity plays in redox homeostasis and impacts vascular function.

## 4. Materials and Methods

### 4.1. Experimental Animals

All animals were used in accordance with our animal protocol, which was reviewed and approved by the University of Wisconsin-Madison Animal Care and Use Committee. Experiments were carried out in accordance with the Association for Research in Vision and Ophthalmology Statement for the Use of Animals in Ophthalmic and Vision Research. C57BL/6J (*Cyp1b1^+/+^*) mice and Cyp1b1-deficient (*Cyp1b1^−/−^*) mice were housed and cared for in our animal facility at the University of Wisconsin-Madison. OIR in mice was performed as previously described [92]. Data obtained from male and female mice were analyzed separately.

### 4.2. Cells

Retinal ECs were isolated from immorto-*Cyp1b1^+/+^* and -*Cyp1b1^−/−^* mice, as previously described [14]. Cells were cultured in Dulbecco’s modified Eagle’s medium (DMEM, D-5523; Sigma, St. Louis, MO, USA) containing 10% fetal bovine serum (26140-079; Gibco, Grand Island, NY, USA), 2 mM L-glutamine (25030-081; Gibco), 2 mM sodium pyruvate (11360-070; Gibco), 20 mM HEPES (15630-080; Gibco), 1% non-essential amino acids (11140-050; Gibco), 100 μg/mL streptomycin, 100 U/mL penicillin (15140-122; Gibco), 55 U/mL heparin (H3149; Sigma), 100 μg/mL endothelial growth supplement at (E2759; Sigma), and murine recombinant interferon-γ (485-MI-100; R&D, Minneapolis, MN, USA) at 44 U/mL. Cells were cultured in 60 mm culture plates (12556001; Thermo Fisher, Hanover Park, IL, USA) coated with 1% gelatin (G1890; Sigma) in phosphate-buffered saline (PBS, D1408; Sigma) and maintained at 33 °C with 5% CO_2_. These cells express a temperature-sensitive large T antigen, and expression is induced by interferon γ, allowing the cells to readily proliferate when cultured at 33 °C. Cells used in these studies were used before passage 15. To observe the effect of iron chelators on cellular morphologies of confluent EC layer, ~90% confluent cells were incubated with deferoxamine (DFO, D9533; Sigma) or 2,2′-Bipyridyl (Bip, D216305, Sigma), or deferiprone (DFP, 379409; Sigma) for 48 h and photographed using a phase contrast microscope (TS100; Nikon, Japan). 

### 4.3. RNA Isolation, cDNA Preparation, and Real-Time Quantitative PCR (qPCR) Analysis

Total RNA was isolated using Trizol reagent (15596026; Life Technologies, Grand Island, NY, USA) and RNeasy Mini Kit (74104; Qiagen, Maryland, CA, USA) for purification. Thus, 1 µg of total RNA was used to synthesize cDNA by using the RNA to cDNA EcoDry Premix (Double Primed) (639549; Clontech, Mountain View, CA, USA). Next, 10-fold dilutions of cDNA samples were used as the templates for qPCR assays using the TB-Green Advantage qPCR Premix (639676; Clontech), performed in triplicate on a Mastercycler Real plex (Eppendorf, Enfield, CT, USA). The thermal cycling conditions were as follows: 95 °C for 2 min followed by 40 cycles of amplification (95 °C for 15 s, 60 °C for 40 s) and the dissociation curve step (95 °C for 15 s, 60 °C for 15 s, 95 °C for 15 s). Standard curves were prepared from known quantities for standard target gene with linearized plasmid DNA. The linear regression line for DNA was determined from relative fluorescent units (RFUs) at a threshold fluorescence (Ct). Target genes from cell extracts were quantified by comparing the RFUs at the Ct to the standard curve and normalized by the simultaneous amplification of Rpl13a, a housekeeping gene. The primers used in this study are as follows: Bmp6 forward: 5′-GTGACACCGCAGCACAAC-3′, Bmp6 reverse: 5′-TCGTAAGGGCCGTCTCTG-3′, Slc40a forward: 5′-ATGTGAACAAGAGCCCACCT-3′, Slc40a reverse: 5′-CCCATCCATCTCGGAAAGT-3′, Esrra forward: 5′-CCTTCCCTGCTGGACCTC-3′, Esrra reserve: 5′-CGACACCAGAGCGTTCACT-3′, Hamp forward: 5′-GCATCTTCTGCTGTAAATGCTG-3′, Hamp reverse: 5′-TGGCTCTAGGCTATGTTTTGC-3′. 

### 4.4. Western Blot Analysis

To access the BMP6 signaling pathway in retinal ECs, 5.0 × 10^5^ cells were plated in 60 mm plates and cultured to reach ~70% confluence and harvested as described below. To investigate the effect of estradiol on BMP6 signaling pathway, 3.0 × 10^5^ cells were plated in 60 mm plates using EC culture medium supplemented by charcoal-stripped FBS (F6765; Sigma) and incubated with 1 or 10 nM 17β-estradiol (E2, E2758; Sigma) for 24 h. To demonstrate the effect of iron chelation on NF-κB and eNOS levels in retinal ECs, 3.0 × 10^5^ cells were plated in 60 mm plates and incubated with different concentrations of DFO for 24 h. To determine the impact of iron chelation on HIF-1α stability, 3.0 × 10^5^ cells were plated in 60 mm plates an incubated with 50 µM of DFO for 24 h. To assess the effect of iron chelation on VE-cadherin expression in confluent retinal ECs, 5.0 × 10^5^ cells were plated in 60 mm plates to reach ~90% confluence and incubated with 10 or 20 µM DFO for 48 h after the treatment; then, cells were collected, as described below. 

To prepare cell lysates, cells were rinsed with PBS once and lysed with lysis buffer (10 mM Tris-HCl, pH 7.6, 1 mM EDTA, 1% Triton X-100, 1% Nonidet P-40, 0.1% SDS) containing protease inhibitor cocktail (11836153001; Roche Biochemicals, Mannheim, Germany), 1 mM Na_3_VO_4_, and 1 mM NaF. Protein concentrations were measured using the Pierce Bicinchoninic Acid Protein Assay Kit (23225; Thermo Fisher); 25 µg of samples was mixed with appropriate volumes of 6x SDS sample buffer and separated by SDS-PAGE using 4–20% Tris-Glycine gel (XP04202; Invitrogen). Separated proteins were transferred to nitrocellulose membranes (10600001; Cytiva, Marlborough, MA, USA). Membranes were blocked with skim milk dissolved in Tris-Buffered Saline containing 0.05% Tween-20 (TBST, BP337-500; Thermo Fisher) for 1 h at room temperature. Then, the membranes were incubated with primary antibodies overnight at 4 °C. Primary antibodies used were as follows: anti-BMP6 (ab155693; Abcam, Waltham, MA, USA), anti-BMPR2 (612292; BD Science), anti-Phosphorylated-SMAD1 (9511; Cell Signaling, Danvers, MA, USA), anti-SMAD1 (9743; Cell Signaling), anti-phosphorylated-NF-κB p65 (3033; Cell Signaling), anti-NF-κB p65 (8242; Cell Signaling), anti-eNOS (SCC-654, Santa Cruz, Dallas, TX, USA), anti-HIF-1α (14179; Cell Signaling), and anti-VE cadherin (550548; BD Bioscience). The membranes were washed with TBST and incubated with appropriate horseradish peroxidase-conjugated secondary antibodies (1:5000, Jackson ImmunoResearch, West Grove, PA, USA) for 1 h at room temperature. The protein bands were visualized by enhanced chemiluminescence (ECL) detection using ECL Western Blotting Detection Reagents (RPN2209; Cytiva) and the UVP BioSpectrum 810 imaging system (Thermo Fisher). The same blot was re-probed with a monoclonal antibody to β-actin (MA5-15739; Thermo Fisher), which was used as a loading control. For quantitative analysis, the band intensities were analyzed by ImageJ software version 1.52j (NIH, Maryland, MD, USA).

### 4.5. Indirect Immunofluorescence Staining

Cells (1 × 10^5^) were plated on 22 × 22 mm glass coverslips coated with 5 µg/mL of fibronectin (354008; Corning, Steuben, NY, USA) in DMEM. To determine the effects of iron chelation on lipid peroxidation products 4-hydroxynonenal (HNE) and acrolein levels in retinal ECs, 5.0 × 10^4^ cells were incubated with 50 µM or 10 µM DFO (Sigma) for 24 h. To demonstrate the effects of iron chelation in VE-cadherin localization in retinal EC layer, 5.0 × 10^5^ cells were incubated with iron chelator DFO (10 or 20 µM) for 48 h. The cells were fixed with 4% paraformaldehyde (15710; Electron Microscopy Sciences, Hatfield, PA, USA) in PBS for 15 min on ice. The cells were then rinsed with PBS three times and permeabilized with 0.1% Triton X-100 for 10 min at room temperature. After being washed with PBS three times, the cells were blocked with 1% bovine serum albumin (BSA, BP-9703; Thermo Fisher) in TBS for 1 h. The cells were then incubated with primary antibodies overnight at 4 °C. The primary antibodies used are as follows: anti-estrogen receptor α (06-935; Millipore), anti-Phosphorylated-SMAD1 (Cell Signaling), anti-SMAD1 (Cell Signaling), anti-VE-cadherin (BD Biosciences), anti-4-hydroxynonenal (HNE11-S; Alpha diagnostics, Burlington, NC), and anti-Acrolein (MA5-27553; Invitrogen). The cells were washed with TBS three times for 5 min and incubated with appropriate fluorescent-dye-conjugated secondary antibodies (1:1000, Jackson ImmunoResearch) for 1 h at room temperature. The cells were rinsed with TBS three times and incubated with DAPI (1:2000, D1306; Invitrogen) for 5 min. The cells were mounted on glass slides using Fluoromount-G mounting solution (0100-01; SouthernBiotech, Birmingham, AL, USA) and photographed using a Zeiss Fluorescence microscope (Axiophot, Zeiss, Germany) equipped with a digital camera. For quantitative analysis, fluorescence intensities were measured by ImageJ and averaged at least 5 images per group.

### 4.6. Intracellular Iron Level Analysis

Cells (2 × 10^4^) were plated on 4-well chamber slides (PEZGS0416; Millipore) coated with 2 µg/mL fibronectin (CB4008; BD Biosciences). The cells were incubated with 10 µM DFO (Sigma) or equal volume of PBS as a vehicle control for 48 h. Then, the cells were washed with serum-free DMEM three times and incubated with 1 µM FerroOrange (F374-10; Dojindo, Rockville, MD) prepared in serum-free EC culture medium for 30 min in a 37 °C incubator. The cells were photographed with an inverted fluorescence microscope (EVOS FL Digital Inverted Fluorescence microscope; Invitrogen). Mean fluorescence intensities were measured from 5 images per group.

### 4.7. Cell Viability Assay

Cells (5 × 10^3^) were plated on gelatin (1% in PBS)-coated 96-well plates (130188; Thermo Fisher). The next day, cells were incubated with different concentrations of Erastin (0.05–10 µM, E7781, Sigma) for 24 h. To assess the effect of ERα antagonism, 3.0 × 10^3^ cells were incubated with different concentrations of methyl-piperidino-pyrazole (MPP, M7068; Sigma) for 24 h and then further incubated with Erastin for another 24 h. The viability of the cells was assessed by MTS assay (G5421; Promega, Madison, WI, USA) by measuring absorbance at 490 nm using a plate reader (Epoch, BioTek Instruments, Winooski, VT, USA). Samples were prepared in quadruplicate and repeated twice.

### 4.8. Oxygen-Induced Ischemic Retinopathy

Postnatal day-7 (P7) mice with nursing dams were exposed to 75 ± 5% oxygen for 5 days with the incubator temperature maintained at 23 ± 2 °C. Oxygen level was continuously monitored with a ProOx P110 Oxygen Controller (BioSpherix, Parish, NY, USA). The mice were returned to room air for 5 days. To investigate the effect of chemicals on neovascularization, half of the pups were administrated with daily intraperitoneal (IP) injections of the following chemicals from P12 to P16 in separate studies; Deferoxamine (DFO, D-9533; Sigma, 50 mg/kg), Deferiprone (DFP, 379409; Sigma, 50 mg/kg), Ferrostatin-1 (Fer-1, H3149; Sigma, 5 mg/kg), MPP (Sigma, 1 mg/kg), and Ferric ammonium citrate (FAC, F5879; Sigma, 20 mg/kg) diluted in saline (50 µL). The remaining half of the pups were injected with appropriate vehicles. At P17, pups were sacrificed by CO_2_ inhalation for retinal wholemount preparations as described below. The chemical structures of DFO and DFP were drawn using ChemSketch software (ACD/Labs, Toronto, ON, Canada). 

### 4.9. Retinal Wholemount Staining and Quantification of Neovascularization

Eyes were enucleated from mice and fixed in 4% PFA in PBS for 1.5 h at room temperature and kept in methanol at −20 °C for further processing. The eyes were rehydrated in PBS for 1 h at room temperature. Connective tissues and extraocular muscles attached to the back of the eyecup were removed by using scissors and forceps under a microscope. A 28-gauge needle was then used to make a small hole in the ora serrata and the cornea and lens were removed by cutting along the ora serrata using scissors. Retinas were carefully detached from the eyecups and blocked with a blocking buffer (50% FBS, 20% goat serum (GS), and 0.1% Triton X-100). The retinas were then incubated with anti-collagen IV (AB756P, Millipore) diluted 1:250 in a blocking buffer (20% FBS, 20% GS, and 0.1% Triton X-100) at 4 °C overnight. The retinas were then rinsed with PBS three times for 10 min each and incubated with appropriate fluorescent-conjugated secondary antibody (1:1000, Jackson ImmunoResearch) for 1 h at room temperature. The hyaloid vessels, which were seen with collagen IV staining, were removed using forceps under fluorescence illumination with a stereomicroscope (SMZ25, Nikon, Japan). The retinas were flattened by four radial cuts and mounted on a glass slide with Fluoromount-G mounting solution (0100-01; SouthernBiotech) and photographed by fluorescent microscopy. Vitreous neovascularization on P17 was quantified by using the ImageJ plugin SWIFT_NV developed by Stahl et al. for ImageJ [93] and presented as percentages of the total retinal area, which was measured by ImageJ.

### 4.10. Statistical Analysis

Statistical analysis between two groups was evaluated with Student’s unpaired *t*-test (two-tailed). The differences between groups of three or more were analyzed with a one-way ANOVA with Tukey’s multiple comparison post hoc test using GraphPad Prism version 8.0 (GraphPad Software, San Diego, CA, USA). The mean ± standard deviation is shown. *p* < 0.05 was considered significant.

## Figures and Tables

**Figure 1 ijms-24-02420-f001:**
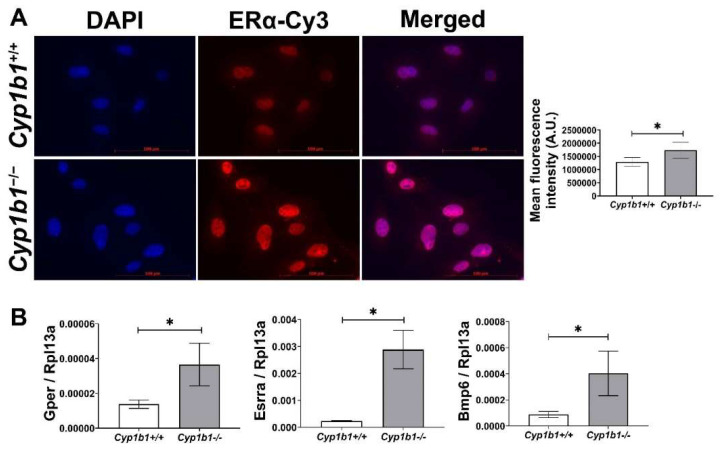
Cytochrome P450 1B1-deficient (*Cyp1b1^−/−^*) retinal ECs exhibit increased ERα nuclear localization and ERα-mediated gene expression. (**A**) The nuclear localization of ERα was determined by indirect immunofluorescence staining. Images were captured in digital format (left panel, scale bars = 100 µm). Integrated fluorescence intensities of images were measured and normalized by cell number for quantitative analysis (right panel). *Cyp1b1^−/−^* retinal ECs showed significantly increased nuclear localization of ERα (* *p* < 0.05; *n* = 5). (**B**) qPCR analysis confirmed enhanced expression of ERα-downstream target genes, including G-protein-coupled ER (Gper), estrogen-related receptor alpha (Esrra), and bone morphogenetic protein 6 (Bmp6) (* *p* < 0.05; *n* = 3).

**Figure 2 ijms-24-02420-f002:**
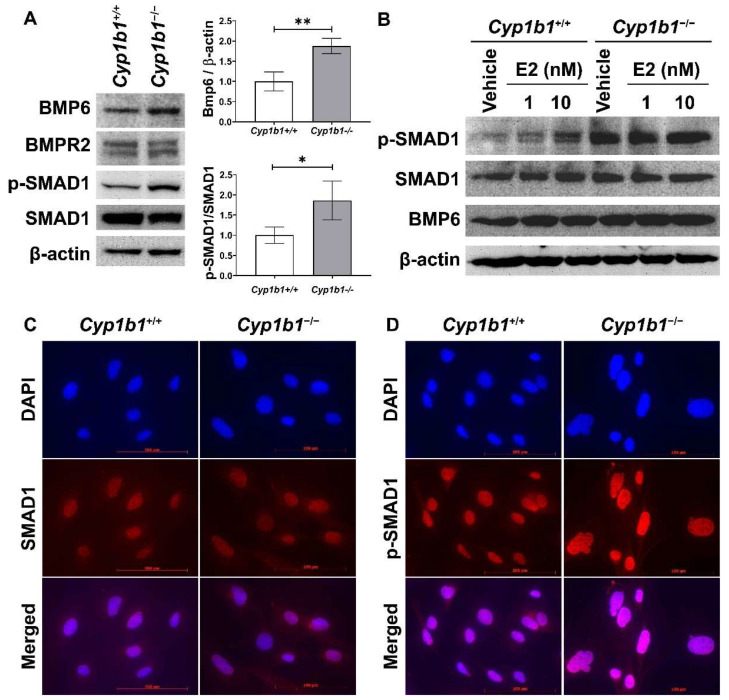
Up-regulation of BMP6 signaling pathway in *Cyp1b1^−/−^* retinal ECs. (**A**) The cell lysates were prepared and used for the analysis of protein levels involved in BMP6 signaling pathway by Western blotting, as detailed in Methods. The levels of BMP6, BMP receptor 2 (BMPR2), p-SMAD1, and SMAD1 proteins were determined using specific antibodies. β-actin levels were determined as a loading control for cell lysates. Band densities were measured from 3 (BMP6/β-actin) and 4 (pSMAD1/SMAD1) blots and analyzed using ImageJ (right panel, * *p* < 0.05, ** *p* < 0.01). (**B**) *Cyp1b1^+/+^* and *Cyp1b1^−/−^* retinal ECs were incubated with 1 nM or 10 nM of 17β-estradiol (E2) for 24 h. The cell lysates were collected for Western blot analysis to determine BMP6, p-SMAD1, and SMAD1 levels. Nuclear localization of SMAD1 (**C**) and p-SMAD1 (**D**) in *Cyp1b1^+/+^* and *Cyp1b1^−/−^* retinal ECs was determined by indirect immunofluorescence staining. SMAD1 and p-SMAD1 were labeled as red and DAPI (blue) was used to stain the nuclei of the cells (scale bars = 100 µm).

**Figure 3 ijms-24-02420-f003:**
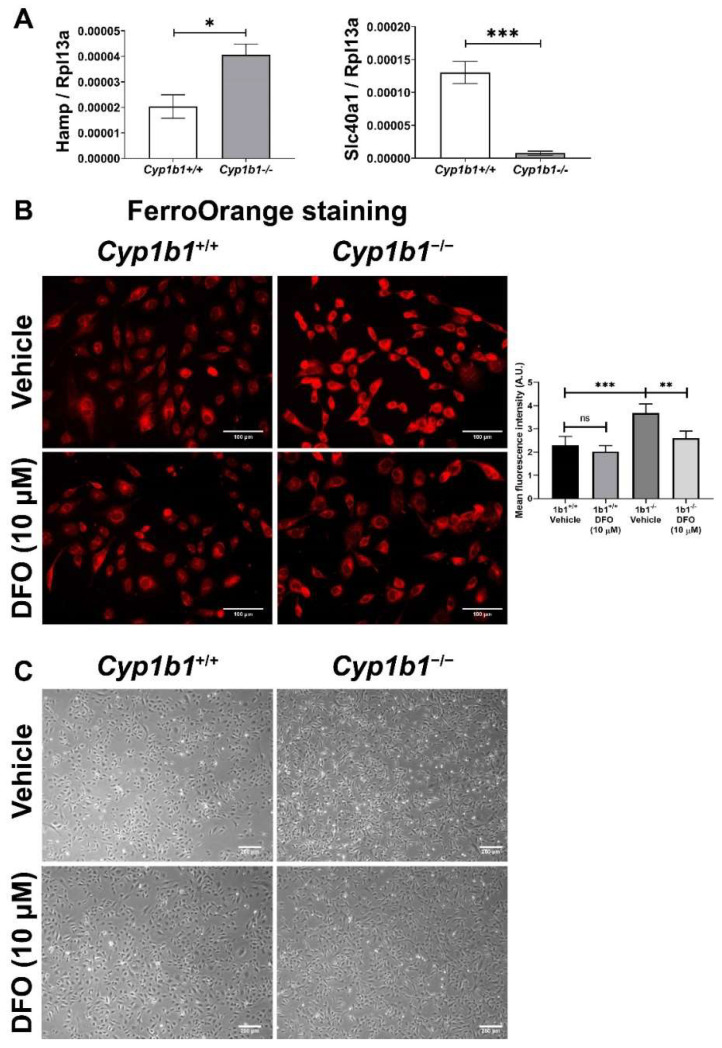
Intracellular iron levels are upregulated in *Cyp1b1^−/−^* retinal ECs. (**A**) The expression levels of Hamp (gene encoding hepcidin) and Slc40a1 (gene encoding ferroportin) in *Cyp1b1^+/+^* and *Cyp1b1^−/−^* retinal ECs were assessed by qPCR analysis (* *p* < 0.05, *** *p* < 0.001; *n* = 3). (**B**) *Cyp1b1^+/+^* and *Cyp1b1^−/−^* retinal ECs were incubated with 10 µM DFO for 48 h. Intracellular iron levels were determined by staining with FerroOrange. The cells were imaged using a fluorescence microscope (scale bars = 100 µm), images captured in a digital format, and mean fluorescence intensities were obtained using ImageJ (** *p* < 0.01, *** *p* < 0.01; *n* = 5; ns: not significant). (**C**) *Cyp1b1^+/+^* and *Cyp1b1^−/−^* retinal ECs were incubated with 10 µM DFO for 24 h and the cells were photographed using a phase microscope (scale bars = 250 µm). Please note iron chelation in *Cyp1b1^−/−^* cells restores a more similar morphology to *Cyp1b1^+/+^* cells.

**Figure 4 ijms-24-02420-f004:**
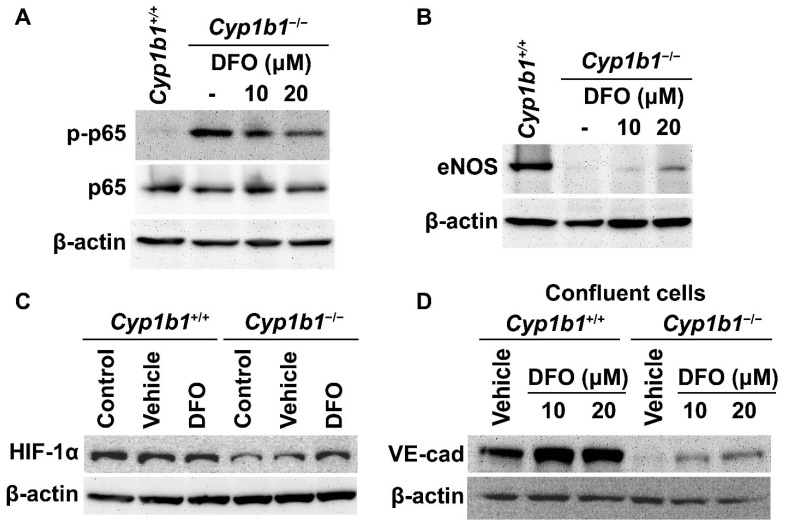
Iron chelation partially reversed molecular changes associated with the lack of CYP1B1 expression in retinal ECs. Cell lysates from *Cyp1b1^−/−^* retinal ECs incubated with 10 or 20 µM of DFO for 24 h and *Cyp1b1^+/+^* retinal ECs without DFO incubation were prepared. Phosphorylated nuclear factor κB p65 (p-p65), p65 (**A**) and endothelial nitric oxide synthase (eNOS) (**B**) levels were determined by Western blot analysis. (**C**) Lysates from *Cyp1b1^+/+^* and *Cyp1b1^−/−^* retinal ECs untreated, incubated with vehicle or 50 µM DFO for 24 h were prepared for Western blot analysis. HIF-1α protein levels were accessed by Western blot analysis. (**D**) Cell lysates prepared from 90% confluent *Cyp1b1^+/+^* and *Cyp1b1^−/−^* retinal ECs incubated with vehicle, 10 or 20 µM DFO for 48 h, were prepared for Western blot analysis. The VE-cadherin (VE-cad) protein levels were determined. The β-actin levels were determined to control for loading.

**Figure 5 ijms-24-02420-f005:**
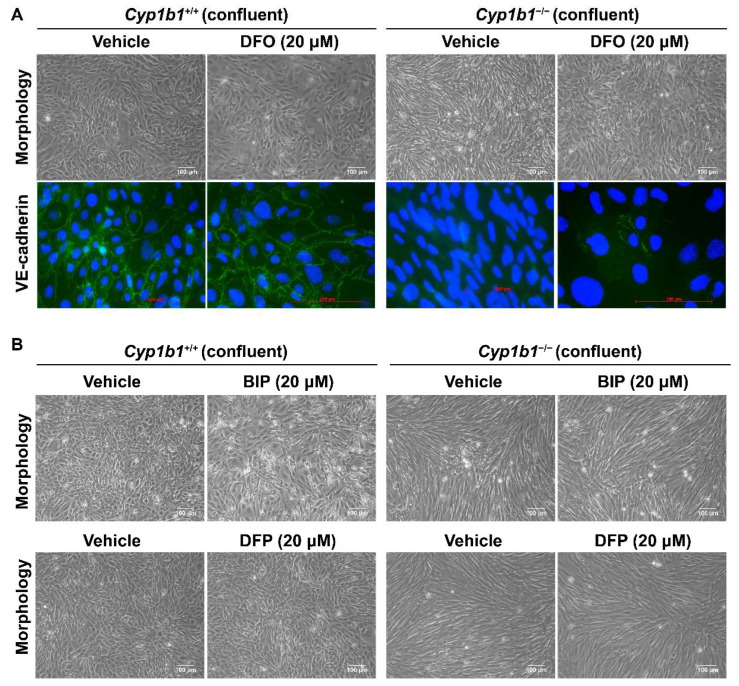
Altered confluent cell morphology and lack of VE-cadherin junctional localization in *Cyp1b1^−/−^* retinal ECs were partially restored by iron chelator DFO, and not by deferiprone (DFP) or 2,2’-Bipyridine (BIP). (**A**) Near-confluent (90%) *Cyp1b1^+/+^* and *Cyp1b1^−/−^* retinal ECs were incubated with 20 µM DFO for 48 h. The cell morphology was assessed by phase microscopy and stained with anti-VE-cadherin antibody (labeled green) to determine the junctional localization of VE cadherin (scale bars = 100 µM). DAPI (blue) was used to stain the cell nuclei. Please note the partial restoration of morphology and VE-cadherin junctional localization in *Cyp1b1^−/−^* retinal ECs incubated with DFO. (**B**) Near-confluent (90%) *Cyp1b1^+/+^* and *Cyp1b1^−/−^* retinal ECs were incubated with 20 µM BIP or DFP for 48 h. The cell morphology was assessed by phase microscopy and photographed in digital format (scale bars = 100 µM). Please note no dramatic changes in morphology of *Cyp1b1^−/−^* retinal ECs incubated with BIP or DFP.

**Figure 6 ijms-24-02420-f006:**
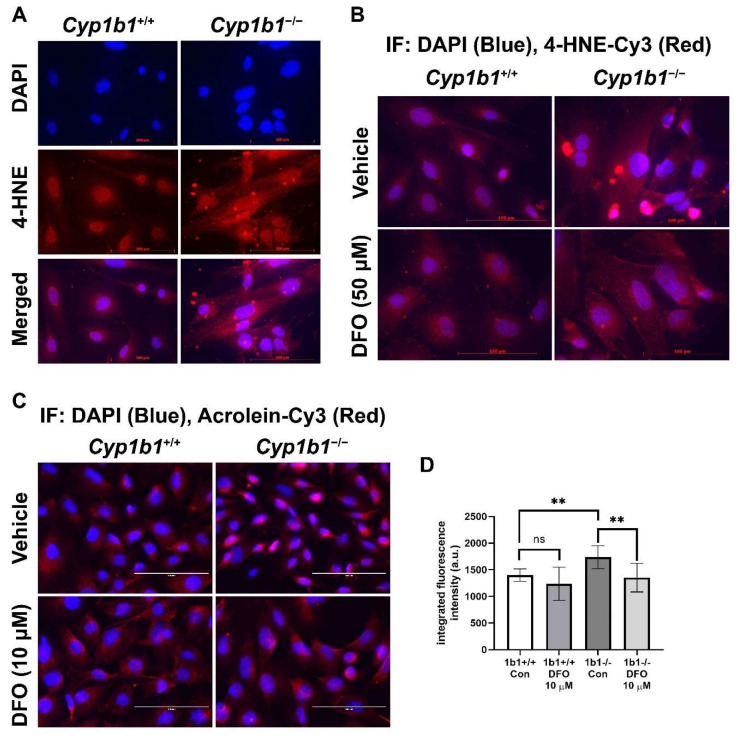
Upregulation of lipid peroxidation in *Cyp1b1^−/−^* retinal ECs. (**A**) *Cyp1b1^+/+^* and *Cyp1b1^−/−^* retinal ECs were stained with anti-4-hydroxynonenal (4-HNE, red) antibody and DAPI (blue) to determine cellular 4-HNE levels (scale bars = 100 µm). (**B**) *Cyp1b1^+/+^* and *Cyp1b1^−/−^* retinal ECs were incubated with 50 µM DFO for 24 h and stained with anti-4-HNE antibody and DAPI. (**C**) *Cyp1b1^+/+^* and *Cyp1b1^−/−^* retinal ECs were incubated with 10 µM DFO for 48 h and stained with anti-acrolein antibody (red) and DAPI (blue) scale bars = 100 µm. (**D**) Acrolein staining intensities were measured using ImageJ for quantitative analysis (** *p* < 0.01; *n* = 10 for each group; ns: not significant).

**Figure 7 ijms-24-02420-f007:**
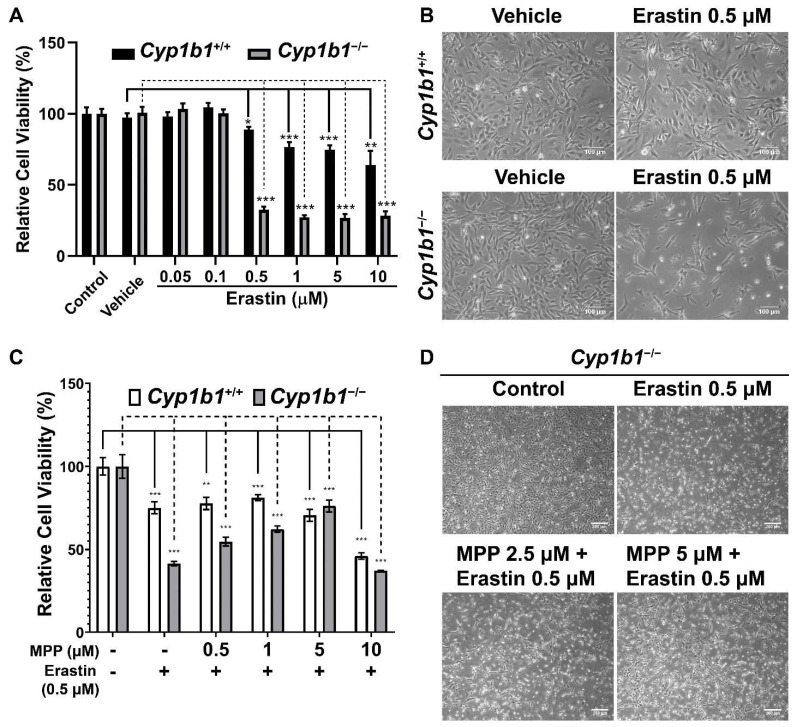
*Cyp1b1^−/−^* retinal ECs are more sensitive to erastin-induced ferroptosis, which is mediated through ERα activity. (**A**) *Cyp1b1^+/+^* and *Cyp1b1^−/−^* retinal ECs were plated on 96-well plates and incubated with different concentrations of erastin for 24 h. Cell viabilities were determined by MTS assay (** *p* < 0.01, *** *p* < 0.001; *n* = 4). (**B**) *Cyp1b1^+/+^* and *Cyp1b1^−/−^* retinal ECs were incubated with 0.5 µM erastin for 24 h. The cells were photographed using a phase microscope (scale bars = 250 µm). (**C**) *Cyp1b1^+/+^* and *Cyp1b1^−/−^* retinal ECs were plated on 96-well plates and incubated with different concentrations of ERα antagonist methyl-piperidino-pyrazole (MPP) for 24 h. The cells were then incubated with 0.5 µM erastin for an additional 24 h. Cell viabilities were determined by performing MTS assay (** *p* < 0.01, *** *p* < 0.001; *n* = 4). (**D**) *Cyp1b1^−/−^* retinal ECs were incubated with MPP (2.5 and 5 µM) for 24 h and then challenged with 0.5 µM erastin for another 24 h. The cells were photographed using a phase microscope (scale bars = 250 µm).

**Figure 8 ijms-24-02420-f008:**
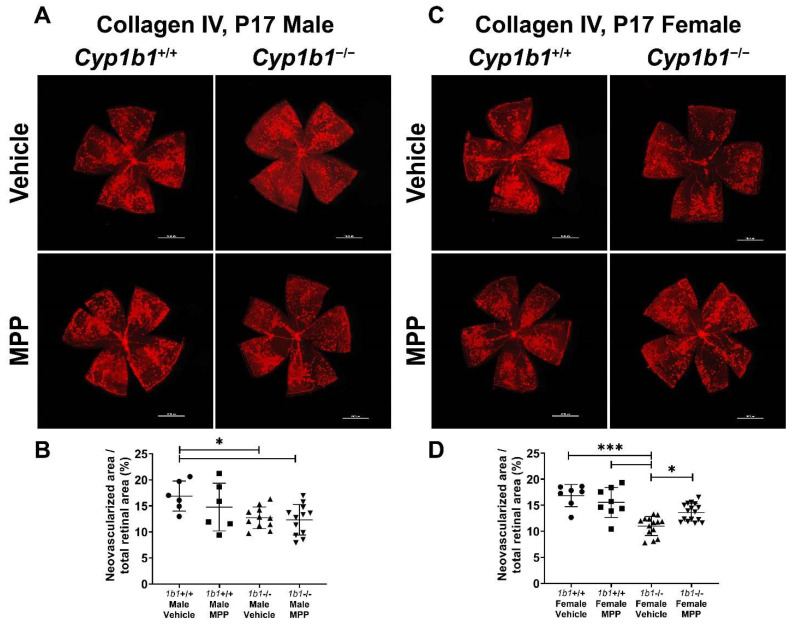
Restoration of retinal neovascularization in female *Cyp1b1^−/−^* mice by ERα antagonist MPP during OIR. Retinas from P17 (**A**) *Cyp1b1^+/+^* mice and (**C**) *Cyp1b1^−/−^* mice that were administrated with MPP (1 mg/kg; P12-P17) during OIR were wholemount stained with anti-collagen IV antibody and imaged by fluorescent microscopy. Scale bars = 1000 µm. Quantitative assessment of the neovascularization was performed using ImageJ and percentages of the neovascularized area relative to whole retina, shown in (**B**,**D**), respectively (* *p* < 0.05, *** *p* < 0.001; *n* ≥ 6; each point represents one retina).

**Figure 9 ijms-24-02420-f009:**
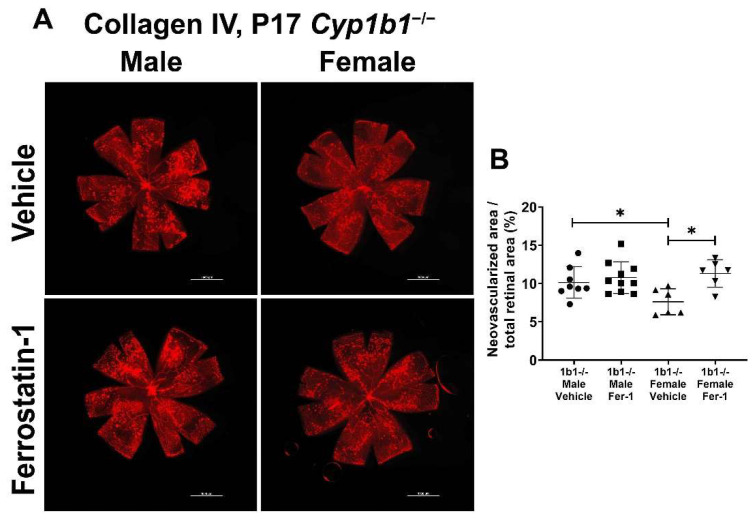
Ferrostatin-1 (Fer-1) administration restored retinal neovascularization in female but not male *Cyp1b1^−/−^* mice during OIR. Retinas from P17 *Cyp1b1^−/−^* mice that were administrated with Fer-1 (5 mg/kg, daily IP injection prepared in 50 µL saline, from P12-P17) during OIR were wholemount stained with anti-collagen IV antibody and imaged by fluorescent microscopy (**A**). Scale bars = 1000 µm. Quantitative assessment of the neovascularization was performed using ImageJ and the percentages of neovascular areas are shown in (**B**). (* *p* < 0.05; *n* ≥ 6; each point represents one retina).

**Figure 10 ijms-24-02420-f010:**
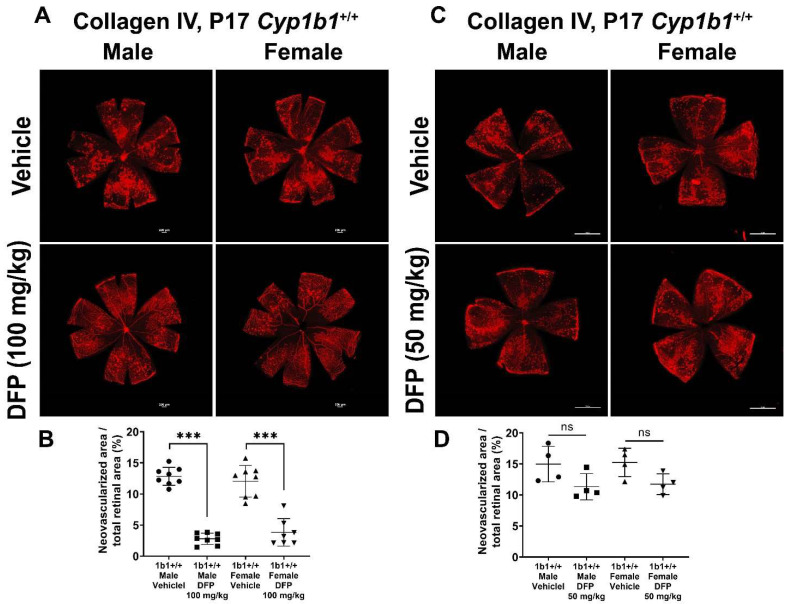
Effect of iron chelation by deferiprone (DFP) on retinal neovascularization in *Cyp1b1^+/+^* mice during OIR. *Cyp1b1^+/+^* mice were administrated with (**A**) 50 mg/kg or (**C**) 100 mg/kg of DFP (daily IP injection prepared in 50 µL saline, from P12 through to P17) in separate OIR experiments. Retinas from those mice at P17 were wholemount stained with anti-collagen IV antibody and imaged by fluorescent microscopy. Scale bars = 1000 µm. Quantitative assessment of the neovascularization was performed using ImageJ and percentages of neovascularized area from these groups are shown in (**B**,**D**) respectively (*** *p* < 0.001, ns: not significant, *n* ≥ 4; each point represents one retina).

**Figure 11 ijms-24-02420-f011:**
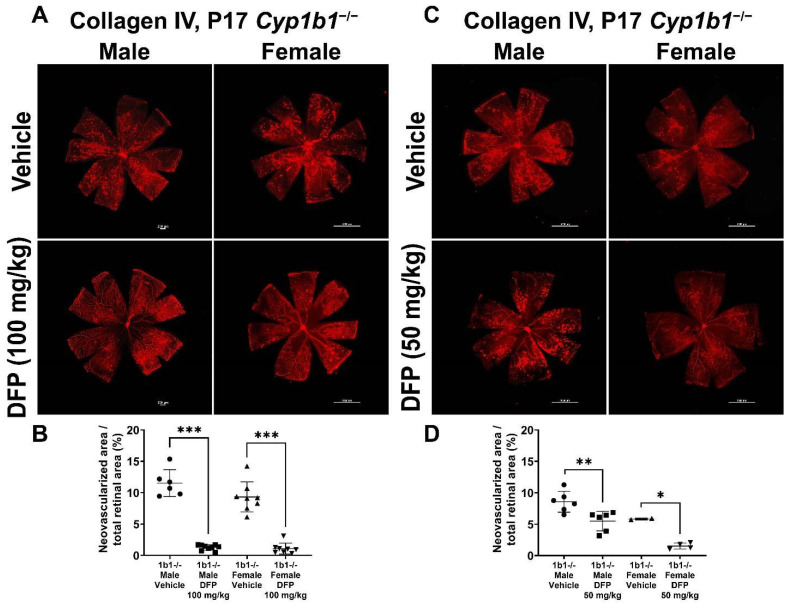
Effect of iron chelation by deferiprone (DFP) on retinal neovascularization in *Cyp1b1^−/−^* mice during OIR. *Cyp1b1^−/−^* mice were administrated with (**A**) 50 mg/kg or (**C**) 100 mg/kg of DFP (daily IP injection prepared in 50 µL saline, from P12 through P17) in separate OIR experiments. Retinas from those mice at P17 were wholemount stained with anti-collagen IV antibody and imaged by fluorescent microscopy. Scale bars = 1000 µm. Quantitative assessment of the neovascularization was performed using ImageJ and percentages of the neovascularized area in the mice are shown in (**B**,**D**) respectively (* *p* < 0.05, ** *p* < 0.01, *** *p* < 0.001, *n* ≥ 2; each point represents one retina).

## Data Availability

All the data presented here are included in the manuscript. Further inquiries should be directed to the corresponding author.

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
