# Peer review of "Cytochrome P450 1B1 Expression Regulates Intracellular Iron Levels and Oxidative Stress in the Retinal Endothelium"

_ijms, 2023, doi:10.3390/ijms24032420_

Round 1
Reviewer 1 Report
The study well designed an the results well supported by Material and Methods.
- Minor spell check is required.
- Line 102 to 122 should be consider in results section.
Author Response
- Minor spell check is required. Thank you. This is done.
- Line 102 to 122 should be consider in results section. This is now moved to discussion (first paragraph).
Reviewer 2 Report
Dear authors,
Thank you for giving me the opportunity to review the article titled Cytochrome P450 1B1 expression regulates intracellular iron levels and oxidative stress in the retinal endothelium.
In this study, the obtained data establish an important role for CYP1B1 expression in the regulation of retinal iron homeostasis and oxidative stress, through modulation of ERα activity and BMP6-hepcidin axis in the retinal endothelium.
The Materials and methods and the obtained results are clearly presented and discussed.
The manuscript is suitable for publication in IJMS, Section Molecular Immunology, Special Issue Molecular Mechanisms of Endothelial Dysfunction 2.0. and could be accepted In the present form.
Author Response
Thank you for giving me the opportunity to review the article titled Cytochrome P450 1B1 expression regulates intracellular iron levels and oxidative stress in the retinal endothelium.
In this study, the obtained data establish an important role for CYP1B1 expression in the regulation of retinal iron homeostasis and oxidative stress, through modulation of ERα activity and BMP6-hepcidin axis in the retinal endothelium. The Materials and methods and the obtained results are clearly presented and discussed.
The manuscript is suitable for publication in IJMS, Section Molecular Immunology, Special Issue Molecular Mechanisms of Endothelial Dysfunction 2.0. and could be accepted in the present form. Thank you for the evaluation and review our manuscript.
Reviewer 3 Report
The research was titled " Cytochrome P450 1B1 expression regulates intracellular iron levels and oxidative stress in the retinal endothelium. This study discussed how CYP1B1 expression in retinal EC regulates intracellular iron levels and significantly impacts ocular redox homeostasis and oxidative stress via modulation of the ER/BMP6/hepcidin axis. Overall, I would recommend the publication in the international journal of molecular sciences. The authors could, please address comments as well as the following minor points:
1. In the introduction, discuss the novelty of this research topic.
2. The confocal images in Figures 1, 2, and 6 appear to be confocal image backgrounds with higher exposure noise levels.
3. In the discussion section, what kind of issue has been identified to emphasize the explanation?
4. The proposed molecular pathway is required to understand how intracellular iron levels are regulated.
5. Revise the references are not uniform format.
Author Response
The research was titled " Cytochrome P450 1B1 expression regulates intracellular iron levels and oxidative stress in the retinal endothelium. This study discussed how CYP1B1 expression in retinal EC regulates intracellular iron levels and significantly impacts ocular redox homeostasis and oxidative stress via modulation of the ER/BMP6/hepcidin axis. Overall, I would recommend the publication in the international journal of molecular sciences. The authors could, please address comments as well as the following minor points:
- In the introduction, discuss the novelty of this research topic. This is now better addressed.
- The confocal images in Figures 1, 2, and 6 appear to be confocal image backgrounds with higher exposure noise levels. These were captured using a regular florescence microscope. We agree more dilution of primary/secondary antibody could have been useful in reducing the background. However, since the levels were different between control and ko cells, in some samples longer exposure was needed, which resulted in creased background in some images.
- In the discussion section, what kind of issue has been identified to emphasize the explanation? This is addressed as suggested.
- The proposed molecular pathway is required to understand how intracellular iron levels are regulated. The pathway discussed here is well established and confirmed by many studies. Therefore, our focus was to demonstrate whether all the key players shown to be important before are affected by CYP1B1 deficiency here.
- Revise the references are not uniform format. We have used the MDPI endnote format and checked for uniformity as suggested.
The manuscript was also evaluated by a native English speaking colleague. As was suggested.